

# Optimizing and purifying extracellular amylase from soil bacteria to inhibit clinical biofilm-forming bacteria

Rokaia Elamary and Wesam M. Salem

Department of Botany and Microbiology, South Valley University, Qena, Egypt

## ABSTRACT

**Background**. Bacterial biofilms have become a major threat to human health. The objective of this study was to isolate amylase-producing bacteria from soil to determine the overall inhibition of certain pathogenic bacterial biofilms.

**Methods**. We used serial dilution and the streaking method to obtain a total of 75 positive amylase isolates. The starch-agar plate method was used to screen the amylolytic activities of these isolates, and we used morphological and biochemical methods to characterize the isolates. Optimal conditions for amylase production and purification using Sephadex G-200 and SDS-PAGE were monitored. We screened these isolates' antagonistic activities and the purified amylase against pathogenic and multi-drug-resistant human bacteria using the agar disk diffusion method. Some standard antibiotics were controlled according to their degree of sensitivity. Finally, we used spectrophotometric methods to screen the antibiofilm 24 and 48 h after application of filtering and purifying enzymes in order to determine its efficacy at human pathogenic bacteria.

**Results**. The isolated *Bacillus* species were *Bacillus megaterium* (26.7%), *Bacillus subtilis* (16%), *Bacillus cereus* (13.3%), *Bacillus thuringiesis* (10.7%), *Bacillus lentus* (10.7%), *Bacillus mycoides* (5.3%), *Bacillus alvei* (5.3%), *Bacillus polymyxa* (4%), *Bacillus circulans* (4%), and *Micrococcus roseus* (4%). Interestingly, all isolates showed a high antagonism to target pathogens. *B. alevi* had the highest recorded activity (48 mm) and *B. polymyxa* had the lowest recorded activity (12 mm) against *Staphylococcus aureus* (MRSA) and *Escherichia coli*, respectively. On the other hand, we detected no antibacterial activity for purified amylase. The supernatant of the isolated amylase-producing bacteria and its purified amylase showed significant inhibition for biofilm: 93.7% and 78.8%, respectively. This suggests that supernatant and purified amylase may be effective for clinical and environmental biofilm control.

**Discussion**. Our results showed that soil bacterial isolates such as *Bacillus sp.* supernatant and its purified amylase are good antibiofilm tools that can inhibit multidrug-resistant former strains. They could be beneficial for pharmaceutical use. While purified amylase was effective as an antibiofilm, the isolated supernatant showed better results.

Corresponding authors
Rokaia Elamary,
roka.elamary_88@yahoo.com
Wesam M. Salem,
wesam.salem@svu.edu.eg

## INTRODUCTION

Bacterial biofilms have increasingly become a serious threat to human health (*Hall-Stoodley, Costerton & Stoodley, 2004*; *Saber et al., 2017*). These substances have a high level of antibiotic resistance and are hosts to immune response stimulants (*Rodrigues et al., 2016*; *Sharma, Misba & Khan, 2019*). They also play an essential role in the pathogenicity of several chronic human infections (*Parsek & Singh, 2003*). Biofilm removal is a particularly difficult task. The principal method for preventing biofilm formation is applying chemicals or antimicrobials, such as chemical biocides, detergents, and surfactants. Biofilm destruction and prevention are effective methods, as are mechanical removal techniques such as shredding, sonication, freezing, and thawing (*De Carvalho, 2007*; *Kalpana, Aarthy & Pandian, 2012*; *Elamary, Albarakaty & Salem, 2020*). However, because the exopolysaccharide biofilm cells are protected (*Kalpana, Aarthy & Pandian, 2012*), it is difficult to completely remove biofilms using these methods. Using enzymes is also a good strategy for biofilm removal because enzymes are rabidly biodegradable and environmentally harmless (*Xavier et al., 2005*). Amylase is a member of the glycosidic hydrolases, which are digestive enzymes that hydrolyze starch glycosidic bonds (*Kaur et al., 2012*). This family also includes maltotriotic glucose, dextrin, and maltose. Amylase has exhibited excellent antibiotic activity against *Pseudomonas aeruginosa* and *Staphylococcus aureus* marine-derived biofilm-forming bacteria (*Vaikundamoorthya et al., 2018*). Soil is the main part of the terrestrial environment, which is compared with aquatic environments with a large association of microorganisms. Among terrestrial bacteria, *Bacillus* sp. is the best source of amylase producers, including *Bacillus subtilis*, *Bacillus cereus,* and *Bacillus polymyxa* (*El-Fallal et al., 2012*; *Dash, Rahman & Sarker, 2015*). *Bacillus* amylase is thermostable, and retains a high pH, osmolarity, and high pressure, which are important for manufacturing (*Islam et al., 2017*). Antibiotics produced by *Bacillus* sp. such as bacitracin, gramicidin S, polymyxin, and tyrotricidin have exhibited great efficacy against gram-positive and gram-negative bacteria (*Perez, Suarez & Castro, 1992*; *Perez, Suarez & Castro, 1993*; *Yilmaza, Sorana & Beyatlib, 2006*). In this study, we identified and isolated *Bacillus* spp. from soil using morphological and biochemical assays. We compared the antimicrobial activity of these isolates against five human pathogenic strains. We optimized and purified the amylase after determining the optimal temperature, pH, incubation period, and starch levels needed for the greatest purification. Finally, we monitored the antibiofilm activity of the filtrate and purified amylase from these isolates.

## MATERIALS & METHODS

### Soil sample collection

We collected 100 soil samples during January 2019 from different sites across the Luxor governorate (Monshaat Al Amari, 25°41′14″N32°41′40″E, 16.2 km), Egypt. Samples were collected in sterile plastic bags under aseptic conditions and were transported to the laboratory (*Reed & Rigney, 1947*). We added 1 gram of soil to 5 ml of tryptic soy broth (Oxoid, Hampshire, United Kingdom), which we modified with 1% starch to make

enrichment broth. Samples were incubated at 37 °C for 24 h. The landowner, Mohamed El sanousy, approved field sampling.

## Screening and isolation of amylase-producing bacteria

Serial dilution techniques are one of the most precise methods for isolating bacteria from soil (*Jamil et al., 2007*; *Rasooli et al., 2008*). We performed serial dilutions up to $10^{-7}$. We aseptically transferred 100 μl from each dilution, which we spread into tryptic soy agar media fortified with 1% starch. The plates were incubated at 37 °C for 24 h to determine the colony-forming unit (CFU)/ml. The plates were then flooded with iodine that turns blue when it reacts to unhydrolyzed starch. If the starch was hydrolyzed, a clear halo zone would appear against a dark blue background around the colonies that produce amylase (*Gupta et al., 2003*; *Abd-Elhahlem et al., 2015*). We further subcultured bacterial isolates to obtain a pure culture and identified isolates using standard morphological techniques based on colony shape, Gram's staining, spore formation, and biochemical characterization (*Cruickshank et al., 1975*; *Collins & Lyne, 1984*; *Koneman et al., 1992*). Isolates were then maintained in a 70% sterilized glycerol stock at −70 °C for further use.

## Selecting isolates for amylase purification

We selected isolates for amylase extraction and purification, as well as for comparing the purified amylase's antibiofilm activity against some human pathogenic bacteria, according to the starch hydrolysis ratio (SHR) that we calculated using the following equation (*Pranay et al., 2019*):

$SHR =$ clear halozone diameter (mm)/colony growth diameter (mm).

Isolates were subcultured on starch agar plates, which were incubated for 24 h at 37 °C. After incubation, the plates were flooded with iodine. Finally, we calculated SHR using the equation above.

## Optimization of amylase production
### Effect of temperature and incubation periods

The starch nutrient medium was prepared and the pH was adjusted to 7.5. We then inoculated the medium with the tested isolates. The culture was allowed to grow on a rotatory shaker (250 revs/min) at temperatures ranging from 15 to 65 °C over 48 h. We took 20 ml from each culture at all temperatures and time intervals (18, 24, 48, 72, 96, and 120 h) and centrifuged them to remove the bacterial cells. Finally, the supernatant was collected to assay the amylase activity (*Nimisha, Moksha & Gangawane, 2019*).

### Effect of pH

We prepared the starch nutrient medium and adjusted the pH to different values (5, 6, 7, 8, 9, and 10). Each isolate was inoculated into a portion of this medium and were grown at 50 °C for 24 h. We then collected 20 ml from each isolate and applied the same treatment as above to determine amylase activity (*Nimisha, Moksha & Gangawane, 2019*).

### Effect of starch concentration

All *Bacillus* isolates were grown on nutrient broth medium with a pH of 9, except *B. subtilis* which was grown at a pH of 7. Different soluble starch quantities were added to fresh

medium to give final concentrations of 0.1, 0.5, 1, 1.5, 2, 2.5, and 3%. We inoculated each isolate in this medium at 50 °C for 24 h to determine their amylase activity (*Nimisha, Moksha & Gangawane, 2019*).

## Determining amylase activity under optimum conditions

The assay mixture contained 2 ml of a solution made up of 1% starch in 50 mM sodium phosphate buffer (pH 7) and 0.1 ml of enzyme solution. After 10 min. of incubation at 40 °C, we stopped the reaction by adding 2 ml of 3,5 dinitrosalicylic acid (DNS) reagent, and heated the tubes at 100 °C for 5 min. The absorbance was measured spectrophotometrically at 540 nm using a blank containing buffer instead of the culture supernatant. We calculated the amount of reduced sugars from a maltose standard curve (*Meyer, Fisher & Bernfeld, 1951*). Protein was determined using *Bradford*'s (*1976*) method.

## Enzyme purification
### Ammonium sulfate precipitation

The crude amylase enzyme was brought to 45% saturation with ammonium sulfate and was kept overnight in a cold room at 4 °C. We removed the precipitate, brought the supernatant to 85% saturation with ammonium sulfate, and centrifuged it at 8,000 rpm for 10 min at 4 °C. After collecting the precipitate during this step, we stored it at 4 °C (*Shinde & Soni, 2014*).

### Dialysis

This step was conducted to exclude the ammonium sulfate remains and to concentrate the enzyme. We used the dialysis tubes, which were previously soaked in 0.1 M phosphate buffer (pH 6.2), for precipitate dialysis. The precipitate was dissolved in 0.1 M phosphate buffer and was dialyzed against the same buffer (*Roe, 2001*).

### DEAE Sephadex G-200

The crude enzyme preparations of the six culture filtrates were applied separately to a column of DEAE-Sephadex G-200. The enzyme was eluted with a linear gradient of sodium chloride (0 –0.4 M) in 200 ml of sodium phosphate buffer (0.05 M and pH 7), the flow rate was adjusted to 1 ml per 1 min., and 200 ml of eluents were collected into 40 tubes (1 × 7 cm) using an automatic circular fraction collector. We determined enzyme activity and protein concentration in each fraction using the described assay method. Fractions of the highest specific activity were pooled together and kept for further study.

### SDS-PAGE

We carried out polyacrylamide gel electrophoresis according to *Laemmli*'s (*1970*) method using 10% polyacrylamide gel. Purified *B. alvei* and *B. cereus* amylase was loaded into wells parallel to the standard protein markers. The protein bands were stained with Coomassie brilliant blue (Sigma, St. Louis, MO, USA). We estimated the enzyme's relative molecular weight by comparing it to molecular mass standard markers (Fermentas, Vilnius, Lithuania).

## Antibacterial activities

### Antagonistic efficacy of the isolated bacteria

We compared the antagonistic efficacy of all isolates against five human pathogenic strains (*Escherichia coli*, *P. aeruginosa*, *S. aureus* (MRSA), *Klebsiella pneumoniae*, and *Acinetobacter baumanii*). Strains were kindly provided by the International Luxor Hospital in Luxor Governorate, Egypt. We performed screening using the disc diffusion method. All bacteria were cultured on TSB modified with 1% starch, adjusted to $OD_{595} = 0.01$, and incubated at 37 °C at 24 h. The isolated bacterial cultures were centrifuged to exclude the cell debris (6,000 rm for 15 min., Biofuge). We then modified 20 ml of TSA with 1% starch, and poured it in a sterile Petri plate (100 mm diameter). We streaked 100 µl of the five tested pathogens on the plates and punched 6-mm wells in the plates using a sterile borer. The wells were then filled with 100 µl of the isolated bacteria filtrate, and the plates were incubated at 37 °C for 24 h. The inhibition zone was measured using a ruler (*Reinheimer, Demkov & Condioti, 1990*). Standard antibiotics were used as the controls according to the Kirby Bauer disk diffusion method (*Bauer, Sherris & Turk, 1966*). The antibiotics were chloramphenicol (C; 30 µg, Oxoid), oxacillin (OX; 1 mcg, Bioanalyse®), vancomycin (VA; 30 mcg, Bioanalyse®), ampicillin/sulbactam (SAM; 10/10 mcg, Bioanalyse®), penicillin G (P; 10 U; Bioanalyse®), erythromycin (E; 15 mcg, Bioanalyse®), sulfamethoxazole/trimethoprim (SXT; 23.75/1.25 µg, BBL™), cefotaxime (CTX; 30 mcg, Bioanalyse®), gentamycin (GM; 10 µg, Bioanalyse®), meropenem (MEM; 10µg, Bioanalyse®), piperacillin (PIP; 100 µg, Bioanalyse®), and piperacillin-tazobactam (PTZ; 100/10 µg, Bioanalyse®). We interpreted the results using the Clinical Laboratory Standard Institute guidelines (*CLSI, 2017*) to determine whether the tested pathogens were resistant, intermediate, or sensitive against the antibiotics.

### Antibacterial activity of purified amylase enzyme from the isolated Bacillus

We placed 100 µl of purified amylase from the selected isolates according to their SHR in the wells of the agar plates inoculated with the target strains. The plates were incubated at 37 °C for 24 h. The halo zone was measured using a ruler.

### Biofilm formation assay

We determined the biofilm formation ability of the tested pathogens (*E. coli*, *P. aeruginosa*, *S. aureus* (MRSA), *K. pneumoniae*, and *A. baumanii*) using 96-well polystyrene plates (*Seper et al., 2011*) and the methods described by *Salem et al. (2015a)* and *Salem et al. (2015b)*: isolates were subcultured on tryptic soy agar for 24 h at 37 °C, suspended in tryptic soy broth, and adjusted to an $OD_{595}$ of 0.02. We placed 130 µl of each adjusted isolate culture in the microtitre plate (U bottom, Sterilin) for 24 and 48 h at 37 °C. After incubation, the wells were washed with distilled water (six times) and were stained with 0.1% crystal violet for 10 min. The wells were then washed again with distilled water (four times) to remove excess stain. Finally, the wells were destained using 210 µl of ethanol 96%, and the $OD_{595}$ was read using an Infinite® F50 Robotic (Ostrich) Microplate Plate to quantify the amount of biofilm.

***Antibiofilm activity of the isolated Bacillus sp. filtrate and its purified amylase enzyme***

We compared the antibiofilm effects of the isolated bacteria filtrate and the purified amylase from the selected isolates against the five human biofilm pathogenic bacteria using the following spectrophotometric methods: a fresh isolate culture was prepared and adjusted to 0.5 McFarland ($10^6$ CFU/ml), and 30 µl (this volume was selected according to a preliminary experiment) of these cultures and purified amylase enzyme were added to 130 µl of the tested pathogens at an $OD_{595}$ of 0.02 after 24 h of incubation at 37 °C to allow biofilm formation. The plates were then incubated for 24 and 48 h and stained with crystal violet. Wells without isolated cultures or amylase served as controls.

## Statistical analysis

The variability degree of the results was expressed in the form of mean ± standard deviation (mean ±SD) based on three independent determinations ($n = 3$). We statistically analyzed the data by one-way ANOVA analysis and compared the control and treatment groups using the least significant difference (LSD) test at 1% (*) levels (*Snedeco & Cochran, 1980*).

# RESULTS AND DISCUSSION

## Screening and isolating amylase-producing bacteria

Microorganisms that produce amylase are generally isolated from soil and other sources (*Fossi, Taveaand & Ndjonenkeu, 2005*). Our study explored the isolation of amylase-producing bacteria from soil using the serial dilution spread plate technique. *Singh & Kumari (2016)* used a similar method by diluting soil samples on starch agar plates and flooding the plates with an iodine solution. The presence of a halo zone around certain colonies indicated amylase production, and a total of 75 bacterial isolates showed a zone of clearance with a starch agar medium. Bacterial isolates were selected according to their amylolytic activity (Table 1). A similar method was also employed by *Magalhaes (2010)*. We further characterized isolates using morphological and biochemical tests shown in Tables 1 and 2. Our results showed that the 75 isolates were comprised of 19 *B. megaterium*, 12 *B. subtilis*, 10 *B. cereus*, eight *B. thuringiesis*, eight *B. lentus*, four *B. mycoides*, four *B. alvei*, three *B. polymyxa*, three *B. circulans,* and three *Micrococcus roseus*. *B. megaterium* had the highest recorded prevalence (26.7%) and *B. circulans* and *Micrococcus roseus* had the lowest (4%). The CFU of the amylase-producing bacteria in our 100 soil samples ranged from $115 \times 10^3$–$198 \times 10^5$ CFU/ml (Table 1).

## Optimizing amylase production

Using the starch hydrolysis rates shown in Table 2, we selected the six isolates with the highest hydrolysis rates, namely *B. alvei, B. thuringiesis, B. megaterium, B. subtilis, B. cerus,* and *B. lentus* with SHRs of 6.0, 5.67, 5.33, 5.0, 4.0, and 3.5 mm, respectively, for amylase purification.

## Effect of temperature and time intervals

All isolates showed maximum amylase production after 24 h. Similar results were obtained by *Singh & Kumari (2016)*, who observed that the highest amylase activity of some *Bacillus*

**Table 1 Prevalence of *Bacillus* species isolated from soil.**

| Isolates[a]/Parameters | No. of isolates[b] | Percentage[c] (%) | CFUml$^{-1}$[d] |
|---|---|---|---|
| *Bacillus megaterium* | 20 | 26.7 | $115 \times 10^3 - 198 \times 10^5$ |
| *Bacillus subtilis* | 12 | 16 | |
| *Bacillus cereus* | 10 | 13.3 | |
| *Bacillus thuringiesis* | 8 | 10.7 | |
| *Bacillus lentus* | 8 | 10.7 | |
| *Bacillus mycoides* | 4 | 5.3 | |
| *Bacillus alvei* | 4 | 5.3 | |
| *Bacillus polymyxa* | 3 | 4 | |
| *Bacillus circulans* | 3 | 4 | |
| *Micrococcus roseus* | 3 | 4 | |
| Total 10 | 75 | 100 | |

**Notes.**
[a] The amylase-producing bacteria isolated from soil.
[b] Number of each isolated type from the total number of the positive isolated sample.
[c] Percentage of each isolate.
[d] Average colony-forming unit of amylase-producing bacteria per ml of 100 g soil samples (highest value to lowest value).

*sp*. occurred at 24 h of incubation and that the activity began to decrease after 48 and 72 h of incubation time (Fig. 1B). *B. megaterium*, *B. subtilis*, and *B. cereus* showed maximum amylase production at 45 °C, while other isolates showed maximum amylase production at 55 °C. *Mohamed, Malki & Kumosani (2009)* similarly reported that some amylase were stable at 40 °C and some at 50 °C (Fig. 1A).

## Effect of pH

All *Bacillus* isolates showed maximum amylase production at a pH of 8, except for *B. subtilis* which maximally produced amylase at a pH of 7. A previous study by *Behal et al. (2006)* found that the optimum pH for amylase production was 8. Another study by *Singh & Kumari (2016)* reported that while amylase activity was recorded at different pH levels from 5 to 10, maximum activity was observed at pH 7 (Fig. 1C).

## Effect of substrate concentration

Our results showed that *B. subtilis* and *B. cereus* had maximum amylase production at 1.5% soluble starch concentration. The remaining isolates showed maximum amylase production at 2.0% soluble starch concetration (Fig. 1D). *Mishra & Behera (2008)* reported that *Bacillus* strains produced the maximum yield of amylase at a starch concentration of 2%.

## Enzyme activity

We purified extracellular amylase from the *Bacillus* isolated from soil to homogeneity using 45–85% ammonium sulfate precipitation and Sephadex G-200 (Fig. 2). As shown in Table 3, the highest amylase activity was found in *B. alvei* (96.02 U/ml), followed by *B. thuringiesis* (88.64 U/ml). *B. megaterium*, *B. subtilis,* and *B. cereus* showed amylase activities of 80.03, 76.0, and 55.9 U/ ml, respectively. *B. lentus* showed the lowest amylase activity of 45.69 U/ml.

**Table 2  Biochemical activities of amylase-producing bacterial isolates and their starch hydrolysis rates.**

| Isolate/Tests[a] | Gram reaction | Motility | Catalase | Egg yolk lecithinase | Nitrate reduction | Vogas proskauer | Citrate utilization | Gelatin hydrolysis | Starch hydrolysis | Indole production |
|---|---|---|---|---|---|---|---|---|---|---|
| Bacillus megaterium | + | + | + | – | – | – | + | + | + | – |
| Bacillus subtilis | + | + | + | – | + | + | + | + | + | – |
| Bacillus cereus | + | + | + | + | + | + | + | – | + | – |
| Bacillus thuringiesis | + | + | + | + | – | + | + | – | + | – |
| Bacillus lentus | + | + | + | – | – | – | – | + | + | – |
| Bacillus mycoides | + | – | + | + | – | + | + | – | + | + |
| Bacillus alvei | + | + | + | – | – | + | – | + | + | + |
| Bacillus polymyxa | + | + | + | – | + | + | – | + | + | – |
| Bacillus circulans | + | + | + | – | + | – | – | + | + | – |
| Micrococcus roseus | + | + | + | – | + | – | + | – | + | – |

| | Starch hydrolysis rate (mm)[b] | | |
|---|---|---|---|
| | Halo zone (mm) | Diameter of colony (mm) | SHR |
| Bacillus megaterium | 16 | 3 | 5.33 |
| Bacillus subtilis | 10 | 2 | 5.0 |
| Bacillus cereus | 12 | 3 | 4.0 |
| Bacillus thuringiesis | 17 | 3 | 5.67 |
| Bacillus lentus | 14 | 4 | 3.5 |
| Bacillus mycoides | 4 | 2 | 2.0 |
| Bacillus alvei | 18 | 3 | 6.0 |
| Bacillus polymyxa | 7 | 5 | 1.4 |
| Bacillus circulans | 16 | 5 | 3.2 |
| Micrococcus roseus | 10 | 5 | 2.0 |

**Notes.**
[a] Morphological and biochemical tests used for identifying isolated bacteria.
+ Positive.
– Negative.
[b] Starch hydrolysis rate.

## SDS-PAGE

After purification, the SDS- PAGE profile showed a single protein band of amylase for each bacteria, confirming that the enzyme has been purified to homogeneity. The molecular weights of *B. alvei* and *B. cereus* were 60 KDa and 43 KDa, respectively (Fig. 3). The molecular weight of *B. alvei* was similar to that of the amylase isolated from *B. subtilis* (56 KDa and 55 KDa, respectively) (*Bano et al., 2011*; *Takkinen et al., 1983*). The molecular weight of *B. cereus* was equal to the molecular weight of amylase from *B. cereus* and *B. subtilis* (42 KDa) (*Annamalai et al., 2011*; *Das, Doley & Mukherjee, 2004*). Previous studies have reported different molecular weights for amylase isolated from *Bacillus sp*. *Lin, Chyau & Hsu (1998)* found that *Bacillus sp*. can produce five different forms of amylase.

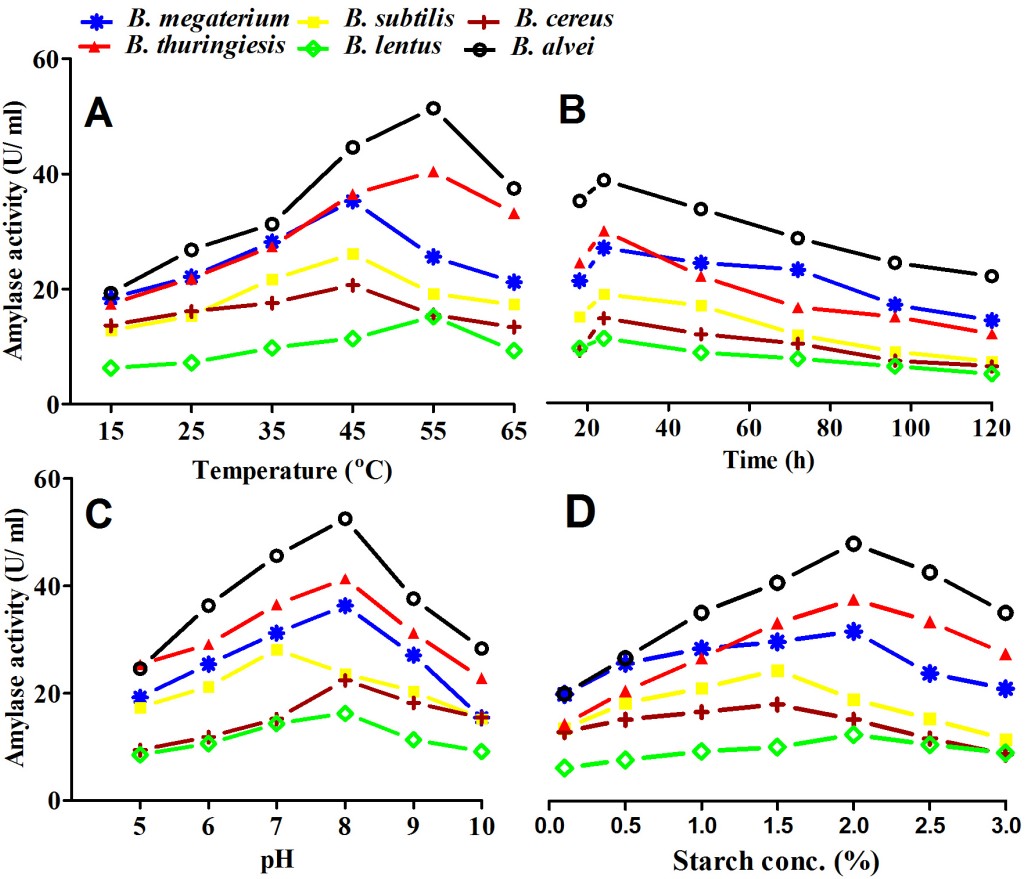

**Figure 1  Optimization and purification conditions of amylase enzyme from selected _Bacillus sp._** (A) The effect of temperature. _B. megaterium_, _B. subtilis_, and _B. cereus_ showed maximum amylase production at 45 °C, while other isolates showed maximum amylase production at 55 °C. (B) The effect of incubation time. All isolates showed maximum amylase production after 24 h incubation. (C) The effect of pH. All _Bacillus_ isolates showed maximum amylase production at a pH of 8.0 except _B. subtilis_, which maximally produced amylase at a pH of 7.0. (D) The effect of starch concentration. _B. subtilis_ and _B. cereus_ had maximum amylase production at 1.5% soluble starch concentration. The remaining isolates had maximum amylase production at 2.0% soluble starch concentration.

## Antibacterial activity
### Antagonistic efficacy of the isolated bacteria and purified amylase enzyme from selected isolates

In this study, we compared the antimicrobial activity of _Bacillus_ supernatant and purified amylase with standard antibiotics against human pathogens (Table 4). The standard antibiotics served as the control group since pathogenic bacteria can become extremely resistant to widely-used antibiotics. The pharmaceutical industry is in need of new and natural antimicrobials that can overcome the problem of multidrug-resistant strains (_Schmidt, 2004_; _Salem et al., 2015a_; _Salem et al., 2015b_; _Salem et al., 2017_). Several soil organisms can produce antibiotics using a survival mechanism that can eliminate their competition (_Talaro & Talaro, 1996_; _Jensen & Wright, 1997_). The _Bacillus_ genus is a terrestrial strain that can produce inhibitory compounds from peptide-derivative and

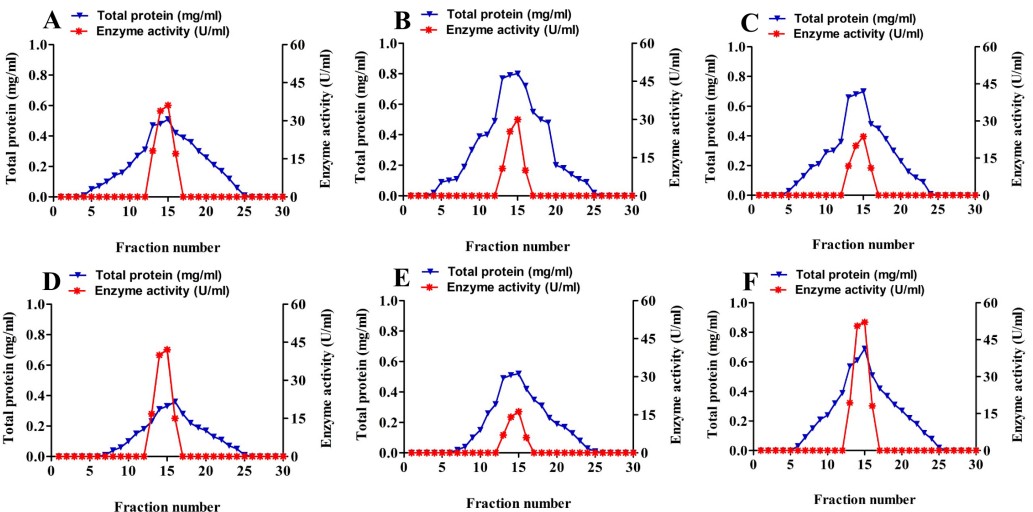

**Figure 2** **Elution profile of isolated *Bacillus sp.* on Sephadex G-200.** The figure shows the total protein concentrations (mg/ml) along with the enzyme activity (U/ml) for (A) *B. megaterium*; (B) *B. subtilis*; (C) *B. cereus*; (D) *B. thuringiesis*; (E) *B. lentus*; and (F) *B. alvei*.

**Table 3** **Purification profile of amylase produced from different *Bacillus* sp. isolates.**

| Isolates[a] Purification step[b] | | Bacillus megaterium | Bacillus subtilis | Bacillus cereus | Bacillus thuringiesis | Bacillus lentus | Bacillus alvei |
|---|---|---|---|---|---|---|---|
| Crude | TP(mg/ml) | 0.66 | 0.96 | 0.81 | 0.54 | 0.99 | 0.86 |
| | TA(U) | 6630 | 5000 | 3932 | 7890 | 2780 | 10040 |
| | EA(U/ml) | 33.15 | 25.0 | 19.66 | 39.45 | 13.9 | 50.2 |
| Ammonium sulfate | TP(mg/ml) | 0.58 | 0.88 | 0.75 | 0.39 | 0.79 | 0.77 |
| | TA(U) | 701.0 | 640 | 461.0 | 820 | 308 | 1029 |
| | EA(U/ml) | 35.06 | 32.0 | 23.08 | 41.0 | 15.4 | 51.47 |
| Dialysis | TP(mg/ml) | 0.50 | 0.82 | 0.71 | 0.37 | 0.60 | 0.70 |
| | TA(U) | 718 | 700 | 502 | 805 | 365 | 591 |
| | EA(U/ml) | 35.9 | 35.0 | 25.12 | 40.26 | 18.26 | 29.56 |
| Sephadex G-200 | TP(mg/ml) | 0.45 | 0.76 | 0.64 | 0.29 | 0.48 | 0.59 |
| | TA(U) | 800 | 760 | 559 | 886 | 456.9 | 960 |
| | EA(U/ml) | 80.03 | 76.0 | 55.9 | 88.64 | 45.69 | 96.02 |

**Notes.**
[a]Isolated *Bacillus* sp. selected for amylase purification according to SHR.
[b]Different purification steps of amylase purification.
 TP, total protein; TA, total activity; EA, enzyme activity.

lipopolypeptide antibiotics (*Mannanov & Sattarova, 2001*; *Tamehiro et al., 2002*; *Stein, 2005*). *Oscariz, Lasa & Pisabarro (1999)* and *Yilmaza, Sorana & Beyatlib (2006)* found that isolated bacteriocin-producing strains such as *Bacillus* sp. were active against gram-negative and gram-positive bacteria. We compared the antimicrobial activity of the isolated amylase-producing bacteria and purified amylase against five human pathogenic bacteria (Table 4). We found that *E. coli* was resistant to sulfamethoxazole-trimethoprim (23.75/1.25 mcg), gentamycin (10 μg), cefotaxime (30 μg), piperacillin (100 μg), and piperacillin-tazobactam
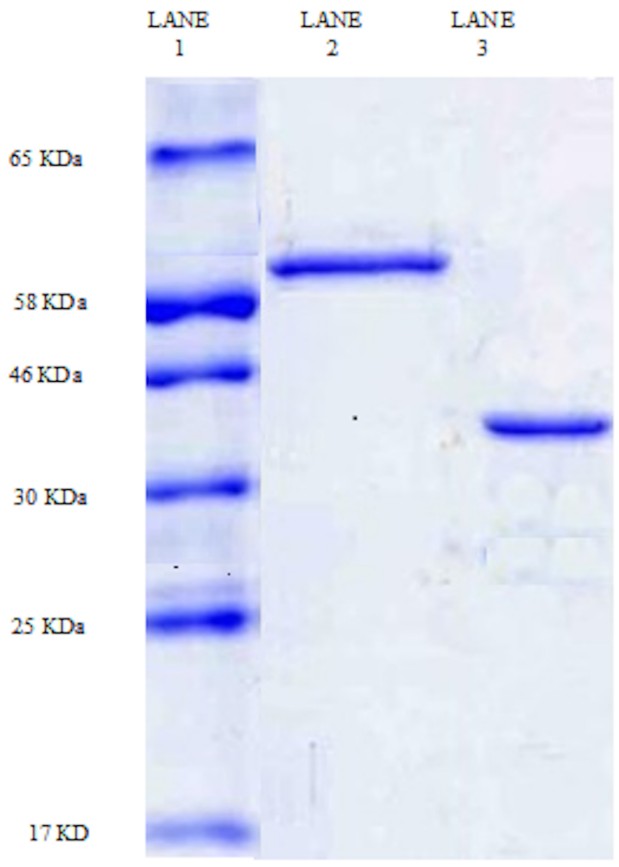

**Figure 3** **SDS-PAGE amylase profile for *B. alvei* and *B. cereus*.** Lane 1: molecular weight marker. Lane 2: amylase purified from *B. alvei*; Lane 3: amylase purified from *B. cereus*.

(100/10 μg); showed intermediate sensitivity to ampicillin-sulbactam (10/10 mcg); and was sensitive to chloramphenicol (30 μg) and meropenem (10 μg). Notably, we observed that all isolated bacteria showed high antimicrobial activity in response to *E. coli,* with *B. polymyxa* showing the most activity (36 mm) and *B. subtilis* and *B. cereus* showing the least (12 mm). *B. mycoides* and *M. roseus* showed no antimicrobial activity in response to *E. coli*. Our results were consistent with the results of *Moshafi et al. (2011)*, who observed that one soil bacterial isolate, identified as *Bacillus* sp., was found to inhibit six pathogenic bacteria, namely *E. coli, K. pneumoniae, S. typhi, P. aeruginosa, S. aureus,* and *S. epidermidis*. When examining the antimicrobial activity in response to *K. pneumoniae*, we found that although *K. pneumoniae* was resistant to all tested antibiotics, it showed intermediate sensitivity to ampicillin-sulbactam (10/10 mcg). It is worth mentioning that all isolates had great antimicrobial effects, with *B. megaterium* showing the highest inhibition (26 mm) and *B. polymyxa* showing the lowest (17 mm) (Table 4). In contrast, *B. mycoides* and *M. roseus* were resistant to *K. pneumoniae*. In a recent study, (*Reed & Rigney, 1947*) reported that *B. subtilis* metabolites inhibited *K. pneumoniae, P. aeruginosa, S. aureus, E. coli, P. mirabilis,* and other bacteria. *A. baumanii* was resistant to all tested antibiotics and was sensitive only

to chloramphenicol (30 μg). However, all isolated bacteria showed improved antibacterial effects against the tested pathogens, with *B. alvei* and *B. cirulans* showing the greatest inhibitory effects (39 mm), and *B. subtilis* and *B. thuringiesis* showing the lowest (21 mm) (Table 4). We found that *B. mycoides* and *Micrococcus roseus* displayed no inhibitory effects against the tested pathogens. *Ramachandran et al. (2014)* reported that *B. subtilis* showed antimicrobial activity against *A. baumanii, E. coli, K. pneumoniae, P. aeruginosa,* and *S. aureus*. The susceptibility level of *P. aeruginosa* indicated that it was resistant to sulfamethoxazole-trimethoprim (23.75/1.25 μg), cefotaxime (30 mcg), gentamycin (10 μg), meropenem (10 μg), and piperacillin (100 μg). It had intermediate sensitivity to chloramphenicol (30 μg) and ampicillin-sulbactam (10/10 mcg), and was sensitive to piperacillin-tazobactam (100/10 μg). We noted that all isolates showed great antibacterial effects against the tested pathogens, with *B. lentus* and *B. cirulans* having the greatest effects (32 mm) and *B. subtilis* having the least (15 mm). *Salem et al. (2015a)* and *Salem et al. (2015b)* similarly reported that *Bacillus* strains exhibited antimicrobial activity against *P. aeruginosa, E. coli,* and *S. typhi. Perez, Suarez & Castro (1992)* and *Aslim, Saglam & Beyatli (2002)* found that *B. subtilis, B. thuringiesis,* and *B. megaterium* showed antibacterial activity against *E. coli* and *P. aeruginosa*. We found that *S. aureus* (MRSA) was resistant to oxacillin (1 mcg), vancomycin (30 mcg), penicillin G (10 U), cefotaxime (30 mcg), and gentamycin (10 μg), and showed intermediate sensitivity to chloramphenicol (30 μg), erythromycin (15 mcg), and sulfamethoxazole-trimethoprim (23.75/1.25 μg). The isolated amylase-producing bacteria showed better antibacterial effects on the tested pathogens, with the greatest effect shown by *B. alvei* (48 mm) and the least effect shown by *B. cereus* (14 mm) (Table 4). However, *B. mycoides* and *M. roseus* did not affect *S. aureus*. Similar results were obtained by *Moshafi et al. (2011)* and *Ramachandran et al. (2014)*. In contrast to the high antimicrobial activity observed in the isolated soil bacteria, the purified amylase from the selected isolates had very little effect on *E. coli* and *K. pneumoniae* (the highest inhibition diameter was 7.5 mm), and no recorded effect in response to the other tested pathogens (Table 4). This result is similar to that of *Kalpana, Aarthy & Pandian (2012),* who confirmed that amylase enzyme has no antibacterial effect.

### Biofilm formation assay

We quantitatively determined the amount of biofilm ($OD_{595}$) in the tested pathogens and designated the 24 and 48 h treatments as the control groups (Figs. 4, 5, 6, and 7). Using the $OD_{595}$ nm mean values, we defined the pathogens as low, moderate, or high bacterial biofilm formers when the $OD_{595}$ nm was > 1, 1 - 2.9, and < 2.9, respectively. *A. baumanii* and *Klebsiella pneumoniae* were high biofilm formers while *E. coli, P. aeruginosa,* and *S. aureus* (MRSA) were low biofilm formers.

### Antibiofilm activity of isolated bacterial filtrate and purified amylase enzyme from selected Bacillus isolates

In a natural ecosystem, bacteria can be exist in two forms: planktonic cells, which are susceptible to antibiotics and other antimicrobial agents, and biofilm, which are resistant to antibiotics and disinfectants (*Limoli, Jones & Wozniak, 2015*). A biofilm is a complex community of bacteria attached to a surface or interface enclosed in an exopolysaccharide

**Table 4  Comparing the antibacterial activity of *Bacillus* sp. and purified amylase enzyme to different standard antibiotics.**

| Tested pathogens/Antibiotic[a]/Isolates | *Escherichia coli* | *Klebsiella pneumoniae* | *Acinetobacter baumanii* | *Pseudomonas aeuroginosa* | *Staphylococcus aureus* (MRSA) |
|---|---|---|---|---|---|
| Chloramphenicol (30 µg) | S | R | S | I | I |
| Oxacillin (1 mcg) | NA | NA | NA | NA | R |
| Vancomycin (30 mcg) | NA | NA | NA | NA | R |
| Ampicillin- sulbactam (10/10 mcg) | I | I | R | I | NA |
| Pencillin G (10 U) | NA | NA | NA | NA | R |
| Erythromycin (15 mcg) | NA | NA | NA | NA | I |
| Sulfameth.-trimethoprim (23.75/1.25 µg) | R | R | R | R | I |
| Cefotaxime (30 mcg) | R | R | R | R | R |
| Gentamycin (10 µg) | R | R | R | R | R |
| Meropenem (10 µg) | S | R | R | R | NA |
| Piperacillin (100 µg) | R | R | R | R | NA |
| Piperacillin-tazobactam (100/10 µg) | R | R | R | S | NA |
| **Inhibition zone in mm[b]** | | | | | |
| *Bacillus megaterium* | 21 ± 1.5 | 26 ± 1 | 36 ± 1 | 24 ± 1 | 31 ± 1 |
| *Bacillus subtilis* | 12 ± 1 | 18 ± 2 | 21 ± 1 | 15 ± 1.5 | 20 ± 2 |
| *Bacillus cereus* | 12 ± 1.2 | 21 ± 1 | 36 ± 1 | 18 ± 1 | 14 ± 1.5 |
| *Bacillus thuringiesis* | 14 ± 1.5 | 21 ± 1 | 21 ± 1 | 22 ± 1 | 18 ± 0.6 |
| *Bacillus lentus* | 22 ± 1.6 | 22 ± 1 | 24 ± 0.6 | 32 ± 1.5 | 31 ± 3 |
| *Bacillus mycoides* | NI | NI | NI | NI | NI |
| *Bacillus alvei* | 34 ± 1.5 | 21 ± 1 | 39 ± 1.5 | 29 ± 2.5 | 48 ± 2 |
| *Bacillus polymyxa* | 36 ± 2.5 | 17 ± 0.6 | 38 ± 1 | 17 ± 4 | 20 ± 0.6 |
| *Bacillus circulans* | 21 ± 1.5 | 23 ± 1 | 39 ± 1 | 32 ± 1 | 32 ± 2 |
| *Micrococcus roseus* | NI | NI | NI | NI | NI |
| **Inhibition zone in mm[c]** | | | | | |
| ABM | 7.2+0.3 | 7.5+0 | NI | NI | NI |
| ABS | 7.3+0.3 | 7+0 | NI | NI | NI |
| ABC | 7+0.3 | 7.3+0.3 | NI | NI | NI |
| ABT | 7.5+0 | 7.2+0.3 | NI | NI | NI |
| ABL | 7.3+0 | 7.5+0 | NI | NI | NI |
| ABA | 7.5+0.5 | 7.3+0.3 | NI | NI | NI |

**Notes.**

[a]Comparing the antimicrobial susceptibility of a group of standard antibiotics (*CLSI, 2017*) against five human pathogenic strains (control).

[b]Antimicrobial activity of the isolated *Bacillus* sp.

R, Resistant; S, sensitive; I, intermediate; NA, Not applicable (for antibiotics that were not specific to the bacterial strains); NI, No inhibition; C, antimicrobial activity of purified amylase from some isolated *Bacillus* sp.; ABM, amylase purified from *Bacillus* megaterium; ABS, amylase purified from *Bacillus subtilis*; ABC, amylase purified from purified *Bacillus cereus*; ABT, amylase purified from *Bacillus thuringiesis*; ABL, amylase purified from *Bacillus lentus*; ABA, amylase purified from *Bacillus alvei*. Values expressed as mean ± SD.

matrix, protected from unfavorable antibiotics, host defenses, or oxidative stresses (*Shakibaie, 2018*). Microbial biofilms have created huge problems in the treatment of both community and hospital infections. Most antimicrobial agents are unable to penetrate biofilm due to its extracellular polymeric substances (EPS), which act as a barrier protecting the bacterial cells within the biofilm. Therefore, we must use compounds that

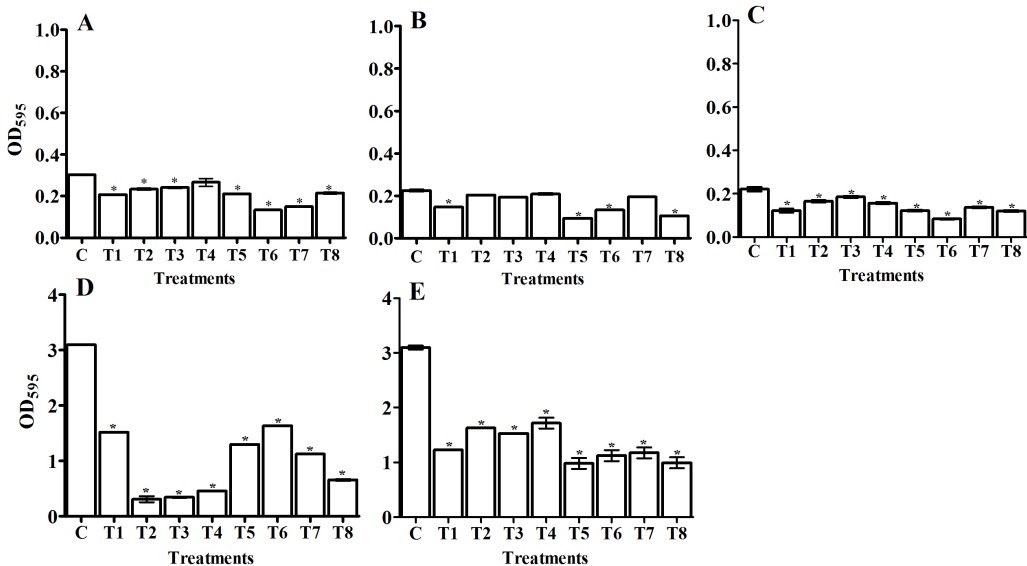

**Figure 4** **Comparing the antibiofilm activity of *Bacillus*-producing amylase-filtrate against some pathogenic bacteria after 24 h treatment.** The figure shows *Bacillus* sp. filtrate, T1: *B. megaterium*, T2: *B. subtilis*, T3: *B. cereus*, T4: *B. thuringiesis*, T5: *B. lentus*, T6: *B. alvei*, T7: *B. polymyxa*, T8: *B. circulans*. The tested pathogenic bacteria are (A) *E. coli*, (B) *P. aeruginosa*, (C) *S. aureus* (MRSA), (D) *K. pneumoniae*, and (E) *A. baumanii*. The figure shows the averages from at least three independent measurements. The error bars indicate the standard deviations using the least significant difference (LSD). The 1% LSD for *E. coli*, *P. aeruginosa, S. aureus* (MRSA), *K. pneumoniae,* and *A. baumanii* was 0.016, 0.013, 0.014, 0.04, and 0.17, respectively. Significant differences between controls and treated samples are marked by asterisks. $P < 0.05$; Krustal-Wallis test and post hoc Bonferroni post-tests. Scales are different for A–C versus D and E.

have the potential to degrade the biofilm's EPS. Enzymes have proven to be effective in EPS degradation (*Kalpana, Aarthy & Pandian, 2012*; *Lequette et al., 2010*). In our study, we compared the antibiofilm activity of the *Bacillus* sp. that we isolated from soil (supernatant) and the purified amylase from these isolates against five human pathogenic biofilm former strains. Our study has reported that *Bacillus* supernatant and amylase enzyme can inhibit the biofilm formation in various pathogens. We confirmed the ability of pathogenic bacterial strains to form biofilm formation using spectrophotometric methods before applying the antibiofilm treatments of bacterial filtrate and purified amylase enzyme (Figs. 4, 5, 6, and 7). The antibiofilm activity was screened using a spectrophotometric method with crystal violet staining. Our results showed that the bacteria isolated from soil exhibited significant antibiolfilm effects against the tested pathogenic strains after 24 h of treatment. The percentage of inhibition significantly increased after 48 h of treatment. The highest percentage of inhibition was recorded for *B. circulans* against *K. pneumonia*: 93.7% after 48 h of treatment (Fig. 5D, T8). We also monitored the efficacy of the purified amylase enzyme as an antibiofilm against the same tested pathogens. Our results revealed that the purified amylase showed significant antibiofilm effects after 24 h of treatment. The percentages of inhibition significantly increased after 48 h of treatment. We observed the highest percentage for *B. alvei* against *K. pneumonia*: 78.8% after 48 h of treatment (Fig.

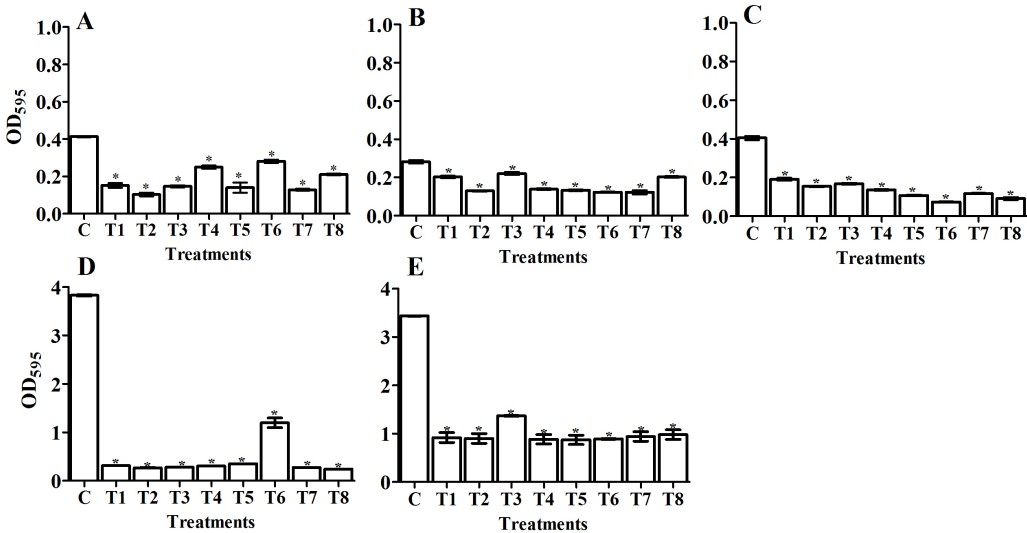

**Figure 5 Comparing the antibiofilm activity of *Bacillus*-producing amylase-filtrate against some pathogenic bacteria after 48 h treatment.** The figure shows *Bacillus* sp. filtrate, T1: *B. megaterium*, T2: *B. subtilis*, T3: *B. cereus*, T4: *B. thuringiesis*, T5: *B. lentus*, T6: *B. alvei*, T7: *B. polymyxa*, T8: *B. circulans*. The tested pathogenic bacteria are (A) *E. coli*, (B) *P. aeruginosa*, (C) *S. aureus* (MRSA), (D) *K. pneumoniae*, and (E) *A. baumanii*. The figure shows the averages from at least three independent measurements. The error bars indicate the standard deviations using the least significant difference (LSD). An LSD of 1% for *E. coli, P. aeruginosa, S. aureus* (MRSA), *K. pneumoniae,* and *A. baumanii* was 0.027, 0.014, 0.013, 0.08 and 0.19, respectively. Significant differences between controls and treated samples are marked by asterisks. $P < 0.05$; Krustal-Wallis test and post hoc Bonferroni post-tests. Scales are different for A–C versus D and E.

7D, T6). Our results also showed the greatest enzyme activity from *B. alvei* (96.02 U/ ml), followed by *B. thuringiesis* (88.64 U/ ml), *B. megaterium* (80.03 U/ ml), and *B. subtilis* (76.0 U/ml). The lowest antibiofilm efficacy was recorded in *B. cereus,* with an enzyme activity of 55.9 U/ml, and *B. lentus,* with an enzyme activity of 45.69 U/ml. *Kalpana, Aarthy & Pandian (2012)* first reported that purified amylase enzyme from *B. subtilis* was a good antibiofilm agent against biofilm-forming clinical pathogens. The purified enzyme caused 68.33%, 64.84%, 61.81%, and 59.2% of inhibition in *V. cholerae* (VC5, VC26), MRSA (102), and *P. aeruginosa* ATCC10145, respectively. Another study by *Vaikundamoorthya et al. (2018)* confirmed the antibiofilm efficacy of the thermostable amylase enzyme from *B. cereus.*

It is worth noting that the isolated bacteria filtrate showed great antibiofilm activity compared to the purified amylase enzyme from the selected isolates. This may be due to the accumulation of some extracellular and intracellular metabolites in the medium, which is further explained by the metabolic overflow theory (*Pinu, Villas-Boas & Aggio, 2017*; *Pinu et al., 2018*; *Horak et al., 2019*). *Bacillus* also showed great efficacy in the production of carbohydrate-active enzymes and bioactive compounds, as well as the secretion of a variety of extracellular metabolites and lytic enzymes (*Abdel-Aziz, 2013*). Additionally, *Bacillus* species are the most efficient at producing peptide antibiotic compounds such as polymyxin, colistin, and circulin (*Katz & Demain, 1997*; *Atanasova-Pancevska et al., 2016*).

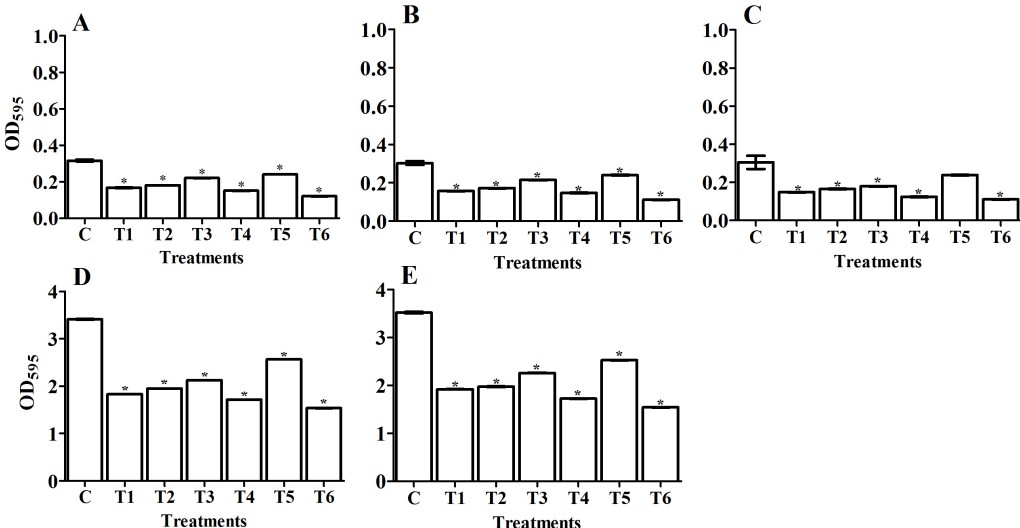

**Figure 6  Comparing the antibiofilm activity of *Bacillus*- purified amylase enzyme against some pathogenic bacteria after 24 h treatment.** The figure shows *Bacillus* sp. filtrate, T1: *B. megaterium*, T2: *B. subtilis*, T3: *B. cereus*, T4: *B. thuringiesis*, T5: *B. lentus*, T6: *B. alvei*. The tested pathogenic bacteria are (A) *E. coli*, (B) *P. aeruginosa*,  (C) *S. aureus* (MRSA), (D) *K. pneumoniae*,  and (E) *A. baumanii*. The figure shows the averages from at least three independent measurements. The error bars indicate the standard deviations using the least significant difference (LSD). An LSD of 1% for *E. coli, P. aeruginosa, S. aureus* (MRSA), *K. pneumoniae,* and *A. baumanii*  was 0.11, 0.01, 0.03, 0.01, and 0.02, respectively. Significant differences between controls and treated samples are marked by asterisks. *P* < 0.05; Krustal-Wallis test and post hoc Bonferroni post-tests. Scales are different for A–C versus D and E.

Our results also indicated a great inhibition of biofilm from the amylase enzymes of *B. alvei* (96.02 U/ml), followed by *B. thuringiesis* (88.64 U/ml), *B. megaterium* (80.03 U/ml), *B. subtilis* (76.0 U/ml), *B. cereus* (55.9 U/ml), and *B. lentus* (45.69 U/ml) (Table 3). This may be due to the increased enzyme activity in each species.

## CONCLUSION

Our results indicated that the ability of *Bacillus sp*. to produce extracellular and intracellular metabolites, lytic enzymes, and some peptide antibiotics directly affects the antimicrobial functions of various *Bacillus sp*. (amylase producers) in the soil. We observed the highest inhibition rate (93.7%) when comparing the species' antibiofilm effects against five human pathogenic strains. We observed an inhibition rate of 78.8% when comparing the antibiotic biofilm activity of purified amylase against the strains. Our study showed that *Bacillus* filtrate is an effective clinical antibiofilm. Futher studies are being conducted to determine the exact composition of the filtrate and its active agents.

### Funding

The authors received no funding for this work.

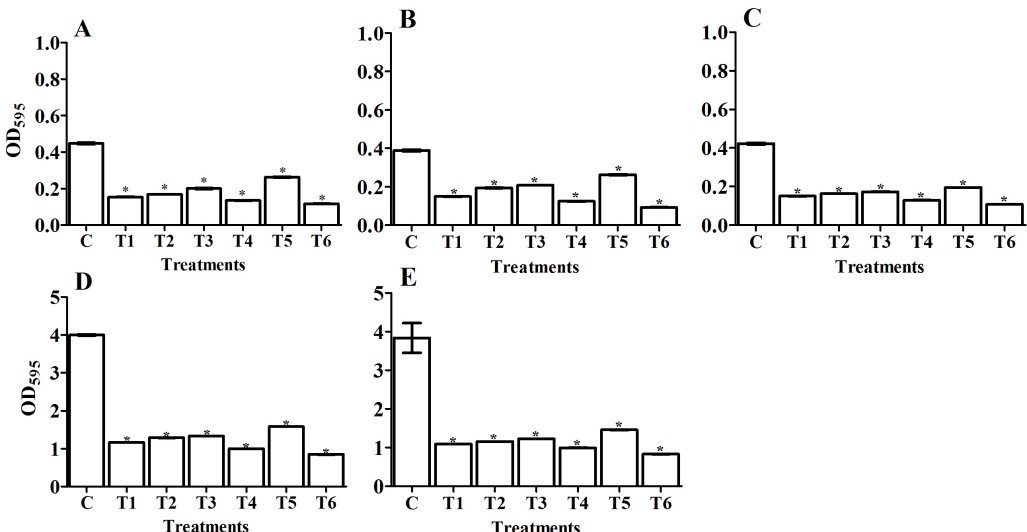

**Figure 7** Comparing the antibiofilm activity of *Bacillus*- purified amylase enzyme against some pathogenic bacteria after 48 h treatment. The figure shows *Bacillus* sp. filtrate, T1: *B. megaterium*, T2: *B. subtilis*, T3: *B. cereus*, T4: *B. thuringiesis*, T5: *B. lentus*, T6: *B. alvei*. The tested pathogenic bacteria are (A) *E. coli*, (B) *P. aeruginosa*, (C) *S. aureus* (MRSA), (D) *K. pneumoniae*, and (E) *A. baumanii*. The figure shows the averages from at least three independent measurements. The error bars indicate the standard deviations using the least significant difference (LSD). An LSD of 1% for *E. coli, P. aeruginosa, S. aureus* (MRSA), *K. pneumoniae,* and *A. baumanii* was 0.01, 0.009, 0.007, 0.018, and 0.35, respectively. Significant differences between controls and treated samples are marked by asterisks. $P < 0.05$; Krustal-Wallis test and post hoc Bonferroni post-tests. Scales are different for A-C versus D and E.

## Competing Interests

The authors declare there are no competing interests.

## Author Contributions

- Rokaia Elamary conceived and designed the experiments, performed the experiments, analyzed the data, prepared figures and/or tables, authored or reviewed drafts of the paper, and approved the final draft.
- Wesam M. Salem conceived and designed the experiments, analyzed the data, prepared figures and/or tables, authored or reviewed drafts of the paper, and approved the final draft.

## Field Study Permissions

The following information was supplied relating to field study approvals (i.e., approving body and any reference numbers):

The landowner, Mohamed El sanousy, approved field sampling.

## Data Availability

Raw data are available as Supplemental Files.

## Supplemental Information

Supplemental information for this article can be found online at http://dx.doi.org/10.7717/peerj.10288#supplemental-information.

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
