# Peer review of "Optimizing and purifying extracellular amylase from soil bacteria to inhibit clinical biofilm-forming bacteria"

_PeerJ, doi:10.7717/peerj.10288_

## Round 0.1 · original submission · Major Revisions

Two reviewers have read and provided insightful comments regarding your manuscript. All agree that a substantial revision is required to improve the manuscript to the point that it is publishable. First and foremost, the manuscript requires significant rewriting for grammar, sentence structure, etc. It was suggested that the authors use a professional editing service or engage a colleague with a strong command of English to proofread and edit the manuscript. In particular, there are many instances where "all" or "every" are used, when in fact, there are exceptions.

Reviewer #1 had some very good suggestions regarding the content of the introduction and discussion. "Relevant previous work related to strategies for biofilm removal and the potential of amylase enzymes in antimicrobial treatments" should be presented and compared against the results reported here. Consider combining the results and discussion sections.

Reviewer #1 also makes a very good point, suggesting that the biofilm inhibition results needs to be revisited. Experimental controls and/or a different approach need to be taken to clearly demonstrate what is being measured - anti-biofilm versus anti-growth activity.

Both reviewers (and the editor) agree Figure 1 could be deleted. Also, the data should be presented as means and standard deviation or standard error, rather than as medians and interquartile ranges.

Finally, reviewer #2 provides some suggestions to improve the presentation of the method such that the experiments could be repeated.

Thank you for your patience during these difficult times...

·

Basic reporting

English use needs to be greatly improved. As it stands now, the manuscript is very hard to follow. I recommend extensive revision by a native speaker or the use of professional editing services.

The Introduction and Discussion sections need significant improvement. The Introduction should include relevant previous work related to strategies for biofilm removal and the potential of amylase enzymes in antimicrobial treatments. Since the focus of this paper is this type of enzymes, previous works where they are used and the rationale of their interest should be better explained. Right now the Introduction is a series of scarcely connected sentences. By the way, biofilms are considered a health threat, but not really a "threat for nature"; they are the usual way in which bacteria (pathogenic or not) are found in nature. As for the Discussion, it should not be repetitive with the Results section. In this particular case it may be even better to combine both.

Figure 1 is of poor quality and not informative, it can be removed. Alternatively, cropped images of the assays with all the isolates and the different target bacteria can be included, with their corresponding labels, but I deem it unnecessary since the quantitative data are shown.

Experimental design

The data from the biofilm inhibition experiments need to be normalized with respect to culture growth. Since antimicrobial activity (that is, growth inhibition) has been observed for some of the isolates on agar plates, it is more than likely that an inhibitory effect also takes place in liquid cultures. This would give the false impression of limited biofilm formation, but of course if bacteria cannot grow they cannot form a biofilm.
I also think biofilm assays should be done in a more exhaustive way in terms of analyzing at different timepoints. 48h seems too long for at least some of the bacteria assayed. Finally, why are the medians presented instead of averages and standard deviations (or standard errors)? This is the most usual way of representing this type of data.

Given the differences in enzyme activity between the purified amylases, I think it is not correct to use the same volume (100 µL, according to the Methods section) of each amylase solution for the plate growth inhibition and biofilm assays. At least, additional assays should be done adjusting the amount used, in order to have approximately the same activity units in all cases. That way one can discern between effects due to higher or lower enzyme activity obtained from each Bacillus strain and those effects related to enzyme specificity towards different bacteria.

Validity of the findings

Without the modifications and additional information indicated above, the validity of some of the findings is questionable.

The Conclusions should reflect (and be limited to) the main findings of the study. In my view these correspond to the antimicrobial potential of several Bacillus isolates, the fact that part of it (but not all) can be due to amylase activity, and the characteristics of the different amylases purified.

Additional comments

The overall concept of the work is scientifically valid and some results are potentially interesting. However, all the aspects indicated above need to be addessed.

Reviewer 2 ·

Basic reporting

The article presents a report of valuable pharmacological activity that can contribute to the development of new therapies against the problem reported in the study.

However, I had some doubts and would like to make some suggestions.

1. Abstract: Something about purification could be mentioned in the methodology, at least the resin you used.

2. Figure 1 "Antagonistic efficacy of isolated Bacillus sp. From soil against some human pathogenic bacteria" is related to some direct result or only indicates the representation of antagonistic efficacy? This figure is mentioned a few times in the results, but it was not clear. In line 230, for example, it was said that figure 1C represents the activity of B. mycoides and M. roseus against E. coli, but figure 1C shows only one plate, where, in the legend, it does not say which bacteria is being represented, moreover, figure 1 does not have a scale and the images are not standardized.

3. Figure 2 "Optimization and purification conditions of amylase enzyme from selected Bacillus sp." could present more information in the legend. There is a point that was not clear: what are the purification conditions in this image?

4. Some minor corrections and suggestions in writing: a) line 61, correct the citation; b) line 80, correct "soli"; line 104: the sentence "The medium inoculated with the tested isolates" is not adequate; line 145, suggest removing the sentence "The used pathogenic strains in the study work".

Experimental design

Materials and methods:

1. About the "Effect of pH", describe the ranges of pH variations used.

2. One of the most important points to ensure that the information can be replicated is to provide the concentration used, especially in antibacterial assays. In line 153, the volume of 100 μl was mentioned, but the concentration in that volume was not mentioned. What concentrations of antibiotics were used? Wouldn't it be interesting to perform a serial dilution with antibiotics / isolates? Inform what is the concentration of "crude enzyme preparations" to use in chromatography.

3. Line 186: what was the temperature used in the incubation?

4. In the methodology, DEAE-Sephadex A-25 resin (line 137) is mentioned, but in table 4 it is Sephadex G-200.

Validity of the findings

Results:

1. Bradford's methodology (1976) was cited, but concentrations were not mentioned.

2. Why hasn't the purification graph (chromatogram) been added? My suggestion at this point is to add the graph and make a polyacrylamide gel (SDS-PAGE) monitoring the purification steps.

3. The topic of discussion can be much more interesting if the effects of dilution are added to the results.

Additional comments

In general, the absence of concentration was what most caught my attention, as it was not mentioned in the methodology, making it difficult to reproduce the experiments.

---

## Round 0.2 · Minor Revisions

Both reviewers agreed that the manuscript is much improved, however several points still need to be addressed before the manuscript is suitable for publication. First, English still needs to be improved throughout the manuscript. Second, the authors should address reviewer #1 concerns about comparing amylase activity across strains as stated, "Normalization with respect to growth is required in this case to support the conclusions. The question of amylase specificity has not been properly addressed, but this may be less relevant, as long as the authors clearly state the growth differences between strains and the fact they are not trying to compare the efficiency of the different amylases but of each amylase vs. total supernatant of the same strain". Finally, consider the suggestions below.

Figure 1. It is somewhat difficult to distinguish the symbols for B. cereus and B. alvei. It would be helpful to change one of these symbols.

Figure 4 legend: Suggested edit for the last sentence. “Significant differences between controls and treated samples are marked by asterisks”

Figures 4-7. Could change the scale of the Y axis for A-C to better visualize differences between bars. The upper value could be 1 instead of 4. Add a note in the legend stating that scales are different for A-C versus D,E.

Table 3. Does ND stand for not done or not detected - as stated in the legend? Not detected does not make sense. Cells are either sensitive, resistant or intermediate.

Lines 330-392. Could certainly decrease text in this section, the percent decrease in biofilm material does not need to be listed for each experiment, as the results are presented in the Figures. Just use the text to highlight the representative, important changes.

·

Basic reporting

The manuscript has been somewhat improved, but in my opinion, the main problems have not been corrected, namely presenting biofilm data normalized with respect to culture growth and adjusting the amount of amylase used to compare differences in specificity. The responses to these issues are not satisfactory enough. The manuscript needs another round of revision in terms of English use and typographic errors.

Experimental design

Same comments as in the first version. I appreciate the addition of an extra timepoint in the biofilm assays, but other aspects have not been satisfactorily responded to. Normalization with respect to growth is required in this case to support the conclusions. The question of amylase specificity has not been properly adressed, but this may be less relevant, as long as the authors clearly state the growth differences between strains and the fact they are not trying to compare efficiency of the different amylases but of each amylase vs. total supernatant of the same strain.

Validity of the findings

no comment

Additional comments

no comment

Reviewer 2 ·

Basic reporting

Standardize the space between the sections in the methodology (lines 215, 221 and 227);

Standardize the left side alignment (lines 363-392).

Experimental design

No comment.

Validity of the findings

No comment.

Additional comments

With the latest changes, the paper makes important contributions to the study of the development of new antibiotics.

---

## Round 0.3 · Minor Revisions

The listing of enzyme activities alongside discussion of the anti-biofilm efficacy of the various amylase extracts is sufficient to meet reviewer #1's concerns in the editors opinion. However, English throughout the manuscript still needs improvement so that clarity of the message is not lost on the readership. PeerJ does offer an English editing service (for a fee) if a strong English editor is not available to the research team.

---

## Round 0.4 · accepted · Accept

Thank you for spending the time/resources to improve the English. The manuscript reads much better and should be well received. Congratulations.